# AF Inducibility Is Related to Conduction Abnormalities at Bachmann’s Bundle

**DOI:** 10.3390/jcm10235536

**Published:** 2021-11-26

**Authors:** Lianne N. van Staveren, Willemijn F. B. van der Does, Annejet Heida, Yannick J. H. J. Taverne, Ad J. J. C. Bogers, Natasja M. S. de Groot

**Affiliations:** 1Department of Cardiology, Erasmus Medical Center, 3015 GD Rotterdam, The Netherlands; l.vanstaveren@erasmusmc.nl (L.N.v.S.); w.vanderdoes@erasmusmc.nl (W.F.B.v.d.D.); a.heida@erasmusmc.nl (A.H.); 2Department of Cardiothoracic Surgery, Erasmus Medical Center, 3015 GD Rotterdam, The Netherlands; y.j.h.j.taverne@erasmusmc.nl (Y.J.H.J.T.); a.j.j.c.bogers@erasmusmc.nl (A.J.J.C.B.)

**Keywords:** Bachmann’s bundle, basic electrophysiology, sinus rhythm, atrial fibrillation, inducibility

## Abstract

We investigated whether patterns of activation at Bachmann’s bundle are related to AF inducibility. Epicardial mapping of Bachmann’s bundle during sinus rhythm was performed prior to cardiac surgery (192 electrodes, interelectrode distances: 2 mm). Compared to non-inducible patients (*N* = 20), patients with inducible AF (*N* = 34) had longer lines of conduction block (18(2–164) mm vs. 6(2–28) mm, *p* = 0.048), prolonged total activation time (55(28–143) ms vs. 46(24–73) ms, *p* = 0.012), multiple wavefronts entering Bachmann’s bundle more frequently (64% vs. 37%, *p* = 0.046) and more often areas of simultaneous activation (conduction velocity > 1.7 m/s, 45% vs. 16%, *p* = 0.038). These observations further support a relation between conduction abnormalities at Bachmann’s bundle and AF inducibility. The next step is to examine whether Bachmann’s bundle activation patterns can also be used to identify patients who will develop AF after cardiac surgery during both short- and long-term follow-up.

## 1. Introduction

A common route for the propagation of the sinus rhythm (SR) wavefront from the right atrium (RA) to the left atrium (LA) is via Bachmann’s bundle (BB). The potential role of Bachmann’s bundle in the pathogenesis of atrial fibrillation (AF) has been described in several clinical studies. Outcomes of these studies suggested that conduction disorders at Bachmann’s bundle may increase the risk of the onset and/or perpetuation of AF [1]. The exact contribution of Bachmann’s bundle to the development of AF is, however, not fully understood [2,3,4].

Interatrial conduction block—defined as a broadened, biphasic p-wave on the surface ECG—was recognized early as a sign of interrupted conduction across Bachmann’s bundle. In the general hospital population, interatrial conduction block was observed in 52% of patients with a history of AF compared to only 18% of patients without prior tachyarrhythmia [2]. In addition, interatrial conduction block is associated with increased susceptibility to new AF onset [5] and increased risk of paroxysmal AF progression to persistent AF [6].

More recently, high-density epicardial mapping studies were performed to investigate conduction disorders across Bachmann’s bundle in more detail. During sinus rhythm (SR), an increased amount of conduction block (CB) and longer lines of CB were found in patients with AF compared to patients without AF [7]. However, it is unknown whether these advanced conduction abnormalities increased AF inducibility.

This study was designed to compare patterns of activation at Bachmann’s bundle between patients with and without inducible AF, using a high-density epicardial mapping approach. If the classification of patterns of activation at Bachmann’s bundle can be used to identify patients with increased AF inducibility, it may also be used to identify patients who will develop AF after cardiac surgery.

## 2. Materials and Methods

This study is part of two prospective observational projects focused on revealing conduction disorders related to the onset and maintenance of AF in patients undergoing cardiac surgery, including the QUASAR project (MEC-2010-054) and Halt&Reverse project (MEC-2014-393), both of which were approved by Erasmus Medical Center’s Medical Ethical Committee. All patients provided written informed consent.

### 2.1. Study Population

Epicardial mapping was performed in adult patients prior to elective open-heart surgery for the correction of ischemic heart disease, valvular heart disease or congenital heart disease. Patients with a pacemaker, assist device, accessory atrioventricular pathways, prior ablative therapy, prior cardiothoracic surgery or left ventricular ejection fraction < 30% were not included. Patient data were selected for the current study if patients had no history of AF, if epicardial SR mapping of Bachmann’s bundle was performed and if subsequent programmed electrical stimulation (PES) resulted in sustained AF (susAF) or non-inducibility (NI). Patient data were collected from digital patient files.

### 2.2. Mapping Procedure

Epicardial mapping was performed after sternotomy and before cardioplegia was induced. A bipolar pacemaker wire was attached to the terminal crest and a steel wire to the thoracic wall in the sternotomy for reference and calibration, respectively. The surgeon used an electrode grid containing 192 electrodes (interelectrode distance: 2 mm) to record 5 s of SR following a predetermined mapping schedule as shown in the upper left panel of Figure 1, including BB. Unipolar epicardial electrograms (U-EGM) were recorded, using a sampling rate of 1000 Hz, a calibration signal of 2 mV and a filter with bandwidth of 0.5–400 Hz.

### 2.3. Induction of Atrial Fibrillation

AF induction was attempted by fixed-rate PES in the RA using a standardized protocol during the mapping procedure. Fixed-rate PES was initiated at a cycle length of 240 ms and decremented until refractoriness was reached. Each burst was applied during 10 s. Patients were categorized according to the duration of induced AF. When the atrial surface was mapped during one single AF episode, it was defined as sustained AF (susAF). The rhythm was considered non-sustained AF when spontaneous termination occurred during the mapping procedure. If PES did not result in susAF or non-sustained AF, patients were assigned to the NI group.

### 2.4. Data Processing

Customized software was used to annotate all atrial signals as previously described in detail [8]. As shown in Figure 1, the steepest negative deflection of each atrial potential was marked as the local activation time (LAT), provided that the signal to noise amplitude ratio was >2. LATs were used to construct color-coded activation maps of each SR beat, as shown in the lower panel. Premature and aberrant atrial beats were excluded from analysis.

### 2.5. Conduction Abnormalities

Color-coded activation maps were used to investigate lines of CB. A line of CB is defined as a difference in LAT (conduction time, CT) between two adjacent electrodes of ≥12 ms, corresponding to a conduction velocity of <18 mm/ms in accordance with definitions applied in previous studies [9]. The proportion of CB is expressed by the number of interelectrode lines of CB relative to the total number of interelectrode connections of the mapping array. The *magnitude of CB* was defined as the size of inter-electrode differences in CT in ms. In addition, color-coded maps were used to assess the classification of the different patterns of activation and total activation time of Bachmann’s bundle (TAT). TAT was calculated as the interval between the first and last LAT during one sinus beat.

### 2.6. Patterns of Activation

In a prior mapping study of Bachmann’s bundle, multiple entry sites of the SR wavefront were observed. Both in patients with and without a history of AF, these entry sites included the RA, the inferior or superior border and center of Bachmann’s bundle [7]. In order to assess the relation between site of entry and AF inducibility, the location of the primary entry site of every SR wavefront was determined. As different parts of the SR wavefront may enter Bachmann’s bundle from multiple sites [7], the number of different wavefront entry sites into Bachmann’s bundle was also taken into account.

The fastest conduction velocity reported at Bachmann’s bundle is 1.7 m/s [10]. As we used a sampling rate of 1000 Hz and interelectrode distance of 2 mm, patterns of activation in which >2 adjacent electrodes were activated within the same time sample were classified as areas of simultaneous activation (SimAct).

### 2.7. Unipolar Electrogram Amplitudes

For each U-EGM, the peak-to-peak amplitude of the steepest deflection was calculated. Peak-to-peak U-EGM amplitudes were compared between patient groups by calculating the relative frequency distribution of amplitudes for each patient separately. Low voltages were defined as U-EGM amplitudes < 1 mV.

### 2.8. Statistical Analysis

IBM SPSS Statistics for Windows, version 25 (IBM Corp., Armonk, NY, USA), was used for all statistical analyses. Normally distributed data were reported as mean (±standard deviation), skewed data as median (range). Two patient groups were compared using Chi square tests for proportions and independent *T-*tests or Mann–Whitney U for continuous data, based on normality. Univariate logistic regression was performed to identify possible predictors of susAF. If variables were significant at a 0.1 level, they were entered in a multivariate regression model that also included LA dilatation (binary variable) and body mass index (BMI)—known independent predictors of AF. Possible overlap between univariate predictors was identified by assessing the change in odds ratios when the variables were combined in a multivariable regression model. Possible correlations between demographics and electrophysiological parameters were investigated using Spearman correlation. Statistical significance was set at 0.05.

## 3. Results

### 3.1. Study Population

Patients without a history of AF (*N* = 103) were enrolled in the Halt&Reverse project (Appendix A). In 77 patients, AF induction either failed (NI: *N* = 20) or resulted in non-sustained AF (*n* = 23) or susAF (*N* = 34). Patients with non-sustained AF were excluded from further analysis, as were two patients with inadequate U-EGM quality. Characteristics of the remaining study population are summarized in Table 1. Left atrial (LA) dilatation—defined as a LA diameter > 4.1 cm in men or >3.9 cm in women—occurred more often in the susAF group (45% vs. 15%, *p* = 0.023).

Table 2 summarizes conduction characteristics in each patient group. Lines of CB were present in all patients except two patients in the susAF group (right panel of Figure 2). However, as illustrated in the left panel of Figure 2, the maximum lengths of lines of CB were longer in the susAF group (maximum length: 18 (2–164) mm versus 8 (2–28) mm, *p* = 0.031). Remarkably, lines of CB longer than 28 mm were only found in the susAF group, occurring in 35% of the patients.

The magnitude of CTs was also larger in the susAF group (magnitude: 17 (12–34) ms versus 14 (12–23) ms, *p* = 0.025). Larger magnitudes were related to longer lines of CB (Figure 2). There were no differences in median lengths of CB lines (susAF: 6 mm (2–22 mm), NI: 4 (2–16) mm, *p* = 0.080), total proportion of CB (susAF: 3.6 (0.0–28.0)%, NI: 1.82 (0.05–7.5)%, *p* = 0.056) or number of CB lines (susAF 3 (0–12), NI: 3 (0–7), *p* = 0.450).

### 3.2. Relation between AF Inducibility and Delayed Activation Time of BB

Boxplots in the right panel of Figure 3 illustrate TAT for each patient in the NI and susAF group. Compared to NI patients, activation time was prolonged in susAF patients (TAT 55 (28–143) ms versus 46 (24–73) ms, *p* = 0.012).

The color-coded activation maps in the left panel of Figure 3 show the increasing complexity of patterns of activation that were observed in three patients with different TATs, ranging from 37 ms to 143 ms. In contrast to the relatively short TAT in the first activation map, large areas of conduction delay resulted in a prolonged TAT in the second and third activation maps (73 ms and 143 ms). The latter activation map corresponds to the longest TAT measured in the entire study population. 

Indeed, TAT was correlated to the maximum length of lines of CB (rho 0.3, *p* = 0.023) and proportion of CB (rho 0.42, *p* = 0.002).

### 3.3. Patterns of Activation

Bachmann’s bundle was activated by more than one wavefront in 21/32 (64%) patients from the susAF group as opposed to only 7/19 (37%) patients in the NI group (*X*^2^ = 4.0, *p* = 0.046). Entry of the SR wavefront into Bachmann’s bundle other than from the RA side—including center, superior or inferior border—did not occur more often in the susAF group (susAF: 22% vs. NI: 33%, respectively).

SimAct areas were observed with variable shapes and sizes in both patient groups (Figure 4), but were more often present in the susAF group, in 14 (42%) versus 3 (15%) patients (*X*^2^ = 4.3, *p* = 0.038).

We found a moderate but significant correlation between the presence of SimAct and BMI (rho 0.40, *p* = 0.003). In addition, SimAct presence was related to an increased amount of conduction abnormalities in Bachmann’s bundle including the proportion of CB (rho 0.58, *p* < 0.0001), maximum length of lines of CB (rho 0.60, *p* < 0.0001) and magnitude of conduction disorders (rho 0.57, *p* < 0.0001).

### 3.4. Unipolar Potential Morphology

Histograms in Figure 5 (left panel) illustrate the relative frequency distributions of peak-to-peak amplitudes of U-EGM for the entire NI (upper panel) and susAF group (lower panel). In the susAF group, U-EGM amplitudes < 1 mV occurred more frequently (12% versus 6%). Likewise, the amount of low voltages per patient was larger (susAF: 6 (0–77)%, NI: 1 (0–38%), *p* = 0.035). This is shown by the scatterplot in the right panel. Patients with the largest amounts of low voltages often had areas of SimAct (indicated by the red markers). Additionally, the proportion of the mapping array displaying SimAct was related to the proportion of low voltages (rho 0.77, *p* < 0.001).

### 3.5. Independent Predictors of AF Inducibility

Binary logistic regression was performed to identify independent predictors of AF inducibility; results of univariate and multivariate analysis are summarized in Appendix A, respectively. LA dilatation predicted AF inducibility (OR 4.7, *p* = 0.030). When corrected for LA dilatation (Appendix A), BMI was a borderline significant predictor of AF inducibility (OR 1.14, *p* = 0.053). Maximum lengths of lines of CB (OR: 1.06, *p* = 0.028), TAT (OR: 1.03, *p* = 0.040) and occurrence of SimAct (OR: 4.2, *p* = 0.026) remained significant predictors of AF inducibility when corrected for LA dilatation only. Only TAT remained a significant predictor of AF induction when corrected for BMI (maximum length of lines of CB: OR 1.16, *p* = 0.056; SimAct presence: OR 3.08, *p* = 0.139; TAT: OR 1.02, *p* = 0.035). A multiple regression model combining the three electrophysiological parameters confirmed that TAT is an independent predictor of AF inducibility, and as for this parameter, the OR remains mostly unaffected by interrelations with the other two parameters (maximum length of lines of CB: OR 1.02, *p* = 0.45; SimAct presence: OR 2.6, *p* = 0.31; TAT: OR 1.05, *p* = 0.065). However, due to a higher number of degrees of freedom in this model (with a control population limited to 20 patients), the predictive value is no longer significant on a 0.05 level (*p* = 0.065).

## 4. Discussion

### 4.1. Key Findings

AF inducibility was related to conduction abnormalities at Bachmann’s bundle including long lines of CB, prolonged activation time of the entire bundle, a higher prevalence of SimAct and parts of SR wavefronts entering Bachmann’s bundle from multiple directions. Lines of CB longer than 28 mm were exclusively present in the susAF group. Areas of SimAct were associated with increased BMI and enhanced conduction abnormalities.

### 4.2. Conduction Abnormalities Related to AF Inducibility

Maximum lines of CB were longer in the susAF group than in the NI group. This is in line with a previous study demonstrating that at BB, longitudinal lines of CB during SR > 10 mm are associated with early postoperative AF [7]. The lengthwise orientation of myocardial strands makes Bachmann’s bundle more sensitive to atrial stretch and may therefore cause a predilection site for conduction abnormalities [11].

The impact of CB at Bachmann’s bundle on AF inducibility remains unclear. A possible explanation is that longitudinally dissociated strands of Bachmann’s bundle may form a core substrate for re-entry circuits. In a canine model, short-coupled extrastimuli at the high LA led to conduction delay between adjacent myocardial strands, splitting Bachmann’s bundle into two functionally separated pathways. As a result, wavefronts evoked at the LA propagated back and forth across Bachmann’s bundle after a single stimulation [12].

In contrast, Bachmann’s bundle may merely serve as a wave passageway and facilitate fast communication between other atrial regions. This was previously observed during acutely induced AF in a sterile pericarditis model, where Bachmann’s bundle was a frequently used route for wavefronts propagating from the septum to the RA and vice versa [13]. Although ‘lines’ of CB are measured in the current study, the longest lines of CB may represent an elaborate framework of clustered, connected lines of CB rather than a long, straight line. In the lower panel of Figure 2, for example, a long line of CB (40 mm) covers an area with moderate diameter (only 8 by 12 mm) but complex configuration. In addition to forming a large re-entry core, these areas of CB may compartmentalize sections of the atria in which re-entry can sustain without interference by other wavefronts. If this is the case, conduction abnormalities at Bachmann’s bundle may play a role in the initiation of AF.

Prior mapping studies demonstrated that enhanced conduction abnormalities at Bachmann’s bundle were not related to an increased amount of conduction disorders in other atrial regions [14,15]; therefore, it is not likely that conduction abnormalities at Bachmann’s bundle are merely a marker of advanced atrial remodeling.

### 4.3. New Kid on the (Conduction) Block

SimAct occurrence was increased almost threefold in patients from the susAF group. To the best of our knowledge, areas of SimAct at Bachmann’s bundle were not reported previously.

One possible explanation for SimAct could be that these areas conduct electrical waves extremely fast and our sampling rate is too low to detect interelectrode differences in local activation time. However, this is very unlikely as conduction velocity would be faster than 1.7 m/s and exceed any previously reported velocities across Bachmann’s bundle. In addition, lines of conduction block would then demarcate lines of very abrupt changes in conduction velocity.

Another explanation may be that broad wavefronts approached the mapping array from the interatrial septum beneath Bachmann’s bundle. However, the areas of SimAct did not resemble emerging wavefronts as the simultaneously occurring electrical activity was not conducted to surrounding tissue.

We cannot exclude that areas of SimAct are the result of remote electrical activity. In this study population, increased BMI was related to areas of SimAct, while in a previous study BMI > 30 kg/m^2^ was related to an increased amount of epicardial fat [16]. Interestingly, in postmortem human atria of 20 patients with and without AF, Becker demonstrated that atrial cardiomyocytes from the terminal crest and Bachmann’s bundle had often disappeared and were replaced by fibro-fatty tissue. Fibro-fatty tissue replacement seemed more extensive in patients with AF; however, no quantitative data were provided [17]. Epicardial fat is also associated with AF, independent of LA dilatation [18].

If fibro-fatty replacement also occurred in our study population, areas of SimAct may be the result of farfield potentials resulting from remote electrical activity in the underlying interatrial septum or cardiomyocytes adjacent to fatty infiltrates [19]. This process may be further enhanced in patients with a higher BMI. If this is the case, lines of CB that demarcate SimAct areas may mark the transition zone from fibro-fatty to muscular tissue.

Fat is a poor conductor; therefore, from our data, we cannot extract whether fatty infiltration has fully replaced cardiomyocytes and annihilated electrical conduction across Bachmann’s bundle. Alternatively, the epicardial adipose tissue may have only formed a thin fatty lining, thereby hampering U-EGM recordings.

### 4.4. Delayed Excitation of BB

In our study population, excitation of Bachmann’s bundle was delayed in patients with susAF. Prolonged TAT of Bachmann’s bundle is caused by either reduced conduction velocity or a prolonged activation pathway.

The slowing of conduction has previously been associated with AF [20]. However, CTs were not prolonged in the susAF group and conduction velocity could therefore not account for the difference in TAT.

Lines of CB lengthen the activation pathway when wavefronts are forced to adjust their trajectory in order to reach the LA. This delay in excitation may also have resulted in the entrance of wavefronts into Bachmann’s bundle from other directions. Wavefronts that enter Bachmann’s bundle at the center of the array, on the other hand, shorten excitation. However, this did not occur more frequently in the NI group.

Delayed excitation also depends on the orientation of lines of CB. Regardless of length, transverse lines of CB may cause a complete interatrial conduction block (IAB). The SR wavefront across Bachmann’s bundle is in this case completely blocked and, instead, propagates from the RA to the LA via the coronary sinus [21]. Subsequently, the activation pathway was prolonged but not related to the length of lines of CB.

Clinical IAB as diagnosed by surface ECG is strongly related to the development of AF [5,22]; however, the relation between prolonged TAT and the clinical diagnosis of IAB is not clear as multiple interatrial conduction pathways have been identified besides Bachmann’s bundle [7]. 

In contrast to our present results, Teuwen et al. found no difference in the TAT of Bachmann’s bundle between patients with and without AF [7]. However, the group of patients included with a history of AF was small (*N* = 13) and patients who showed an activation pattern other than straight right to left propagation were excluded, thus investigating excitation during specific activation patterns rather than delayed excitation in general.

### 4.5. Clinical Relevance of AF Inducibility

Evidence supporting the (long-term) prognostic value of AF inducibility is lacking in patients without structural heart disease. In 52 patients without prior tachyarrhythmia, NI (*N* = 19) was associated with reduced arrhythmia occurrence for up to 2.5 years after the induction attempt. After three years, however, AF onset rate no longer differed between patient groups [23]. In larger study populations, AF inducibility was a poor prognostic biomarker for AF recurrence after ablative therapy [24,25]. Even so, it should be kept in mind that in post-ablative patients, the conductivity of atrial cardiomyocytes may still decrease (scar formation) or recover (reabsorption of local edema) after the procedure. Inducibility in these patients can therefore not be compared to inducibility in the current study population.

As AF inducibility also depends on stimulus strength and the number of induction attempts in patients without structural heart disease [26], we chose to exclude patients with non-sustained AF who may reflect a patient group most vulnerable to variation in pacing protocols.

Due to the low incidence rate of early and late postoperative AF, we could not examine differences in AF onset between susAF and NI patients.

### 4.6. Limitations

As conduction abnormalities were assessed during SR only, potential rate-dependent lines of CB caused by fast rate pacing were not detected. This may have caused underestimation of lines of CB, presumably mainly in the susAF group. Additionally, mapping data acquired in this study do not necessarily account for conduction in deeper tissue layers such as the endocardium. The power of multivariable analyses in this study was limited due to a relatively low number of patients, especially in the NI group. Larger studies should be designed to further analyze the complex interplay of distinct conduction abnormalities that increase susceptibility to AF. Further work including a dedicated study design focusing on Bachmann’s bundle only is required to fully understand the (interaction between) structural and functional conduction abnormalities at Bachmann’s bundle.

## 5. Conclusions

Epicardial high-density mapping during cardiac surgery showed that a prolonged TAT of Bachmann’s bundle, prolonged lines of CB, wavefront entry from multiple sites and areas of SimAct at Bachmann’s bundle are related to increased AF inducibility. In turn, these conduction abnormalities are related to LA dilatation and increased BMI, known indicators of increased susceptibility to AF. These findings support the hypothesis that conduction abnormalities at Bachmann’s bundle are related to AF inducibility. The next step is to examine whether Bachmann’s bundle activation patterns can also be used to identify patients who will develop AF during the short- and long-term follow-up after cardiac surgery and to determine whether Bachmann’s bundle may be a suitable target for AF therapy.

## Figures and Tables

**Figure 1 jcm-10-05536-f001:**
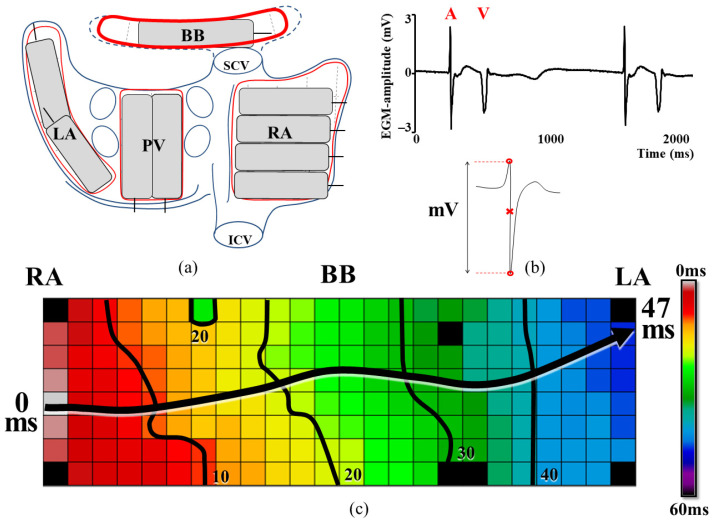
Mapping procedure and signal annotation. (**a**) Epicardial mapping scheme designed to cover the entire atria. (**b**) The steepest negative deflection of each atrial potential, indicated by the red cross, was marked as the LAT. (**c**) Color-coded activation map of a sinus rhythm beat, reconstructed by using LATs. Isochrones are drawn at 10 ms intervals and the arrow indicates the main direction of propagation. A: atrial potential, BB: Bachmann’s bundle, ICV: inferior vena cava, LA: left atrium, LAT: local activation time, PV: pulmonary veins, RA: right atrium, SCV: superior vena cava, V: ventricular far-field potential.

**Figure 2 jcm-10-05536-f002:**
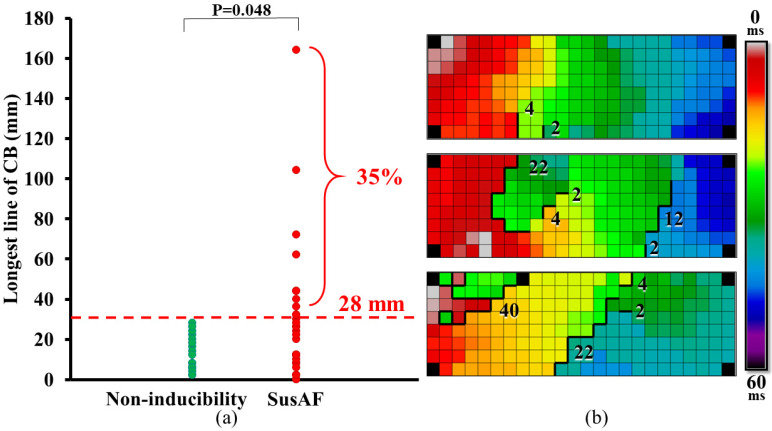
Maximum length of lines of conduction block. (**a**) The longest line of CB for each individual patient in the NI (green markers) and susAF group (red markers). Lines of CB longer than 28 mm were only found in the susAF group, occurring in 35% of the susAF group. (**b**) Three color-coded activation maps in which orientation and length of lines of CB are demonstrated. The upper map shows only 2 short lines of CB (4 and 2 mm, magnitude: each 13 ms), whereas lengths up to 22 mm (magnitude: 27 ms) and 40 mm (magnitude 46 ms) are observed in the middle and lower activation maps, respectively. Thick black lines indicate lines of CB. CB: conduction block, NI: non-inducibility, SusAF: sustained atrial fibrillation.

**Figure 3 jcm-10-05536-f003:**
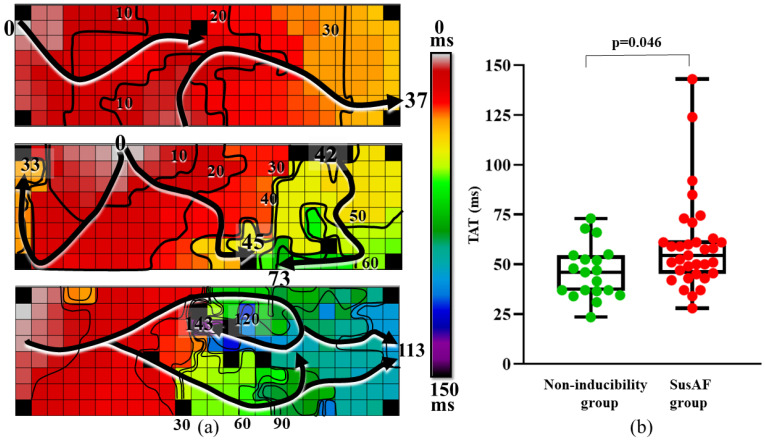
Total activation time of Bachmann’s bundle. (**a**) Increasingly complex patterns of activation observed in three patients with different TATs, ranging from 37 ms (upper) to 143 ms (lower). (**b**) Boxplot of TAT for each individual patient in the NI and susAF groups. Compared to patients from the NI group, excitation time was prolonged in susAF patients (TAT 55 m (28–143 ms) versus 46 ms (24–73 ms), *p* = 0.012). Arrows indicate main wavefront direction, whereas crowding of isochrones, drawn at 10 ms intervals, indicates conduction delay. BB: Bachmann’s bundle, NI: non-inducibility, SusAF: sustained AF, TAT: total activation time.

**Figure 4 jcm-10-05536-f004:**
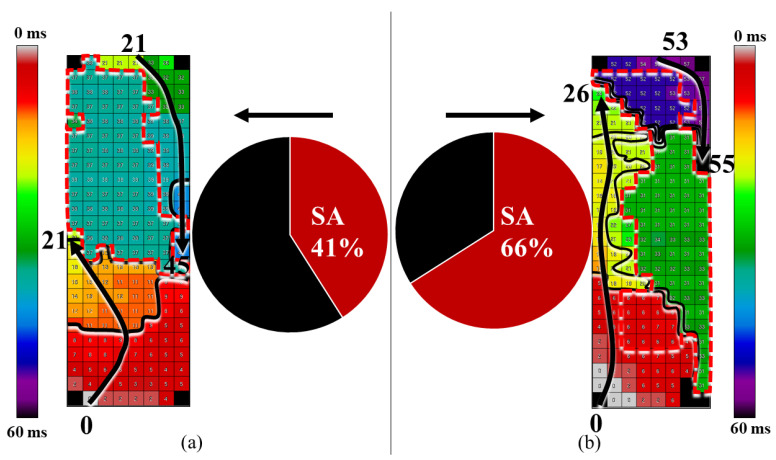
Areas of simultaneous activation. (**a**) Bachmann’s bundle is activated by wavefronts that enter from both the RA and LA side and are extinguished at the perimeter of an area of SimAct (dashed red lines). This SimAct area covers 41% of the mapping area. (**b**) Two SR wavefronts enter the RA and LA side, propagate towards the opposite side of the array along, respectively, the left and right border but remain separated by multiple areas of SimAct, covering 66% of the mapping area. Areas of SimAct are partly delineated by long lines of CB (indicated by crowding of isochrones). Isochrones are drawn at 10 ms intervals and arrows indicate main wavefront direction. CB: conduction block, LA: left atrium, RA: right atrium, SimAct: simultaneous activation, SR: sinus rhythm.

**Figure 5 jcm-10-05536-f005:**
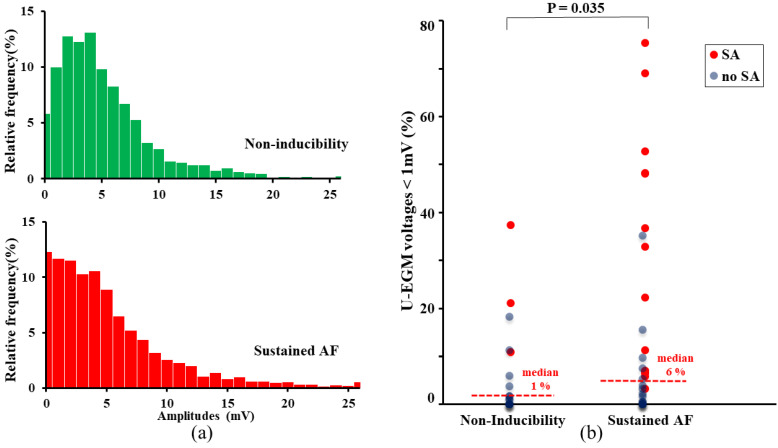
Peak-to-peak amplitudes of the unipolar potentials. (**a**) Relative frequency distributions of peak-to-peak amplitudes of U-EGM for the entire NI and susAF group. U-EGM amplitudes < 1 mV occurred more frequently in the susAF group (12% versus 6%). (**b**) Likewise, the amount of low voltages per patient was larger in the susAF group (susAF: 6 (0–77)%, NI: 1 (0–38%), *p* = 0.035), as illustrated by the scatterplot. NI: non-inducibility, SimAct: simultaneous activation, susAF: sustained atrial fibrillation, U-EGM: unipolar electrocardiogram.

**Table 1 jcm-10-05536-t001:** Patient characteristics.

	**Non-Inducibility (*N* = 20)**	**Sustained AF (*N* = 34)**	** *p* ** **-Value**
Patient characteristics			
Age (years)	62 ± 43	61 ± 15	0.77
Gender (female *N*, (%))	6 (30)	15 (42)	0.37
Risk factors - BMI (kg/m^2^) - Diabetes mellitus, *N*(%) - Hypertension, *N*(%)			
	27 ± 4.0	29 ± 5.0	0.06
	5 (25)	9 (26)	0.73
	9 (45)	18 (53)	0.62
LA enlargement, *N*(%)	3/20 (15)	15/33 (45)	0.023
LV function - Normal - Mild impairment - Moderate impairment			0.70
	18	30	
	2	4	
	0	0	
Underlying heart disease, *N*(%) - CABG - VHD/CABG + VHD - CHD			0.63
	(35)	(35)	
	(40)	(38)	
	(35)	(26)	
Antiarrhythmic drugs - Class I - Class II - Class III - Class IV - Digoxin			
	0	0	0.26
	10 (50)	17 (50)	0.53
	0	1 (3)	0.38
	2 (10)	2 (6)	0.32
	0	0	0.26
Procedural characteristics			
Number of induction attempts	4.6 (±2)	2.0 (±1.3)	<0.001

Relation between AF inducibility and characteristics of conduction disorders.

**Table 2 jcm-10-05536-t002:** Conduction characteristics.

	Non-Inducibility (*N* = 19)	Sustained AF (*N* = 33)	*p*-Value
Conduction times			
- Median CT (ms) - Magnitude of CT (ms)	0 (0–2)	0 (0–2)	0.039
	14 (12–23)	17 (12–34)	0.025
Lines of CB - Median length (mm) - Maximum length (mm) - Total number of lines - Proportion of CB (%)			
	4 (2–16)	6 (2–22)	0.080
	2 (2–28)	18 (2–164)	0.031
	3 (0–7)	3 (0–12)	0.450
	1.8 (0.05–7.5)	3.6 (0.0–28.0)	0.056
Patterns of activation: - TAT (ms) - Simultaneous activation - >1 wave entry site			
	46 (24–73)	55 (28–143)	0.012
	3/20 (15%)	14/34 (41%)	0.038
	7/19 (37%)	21/34 (41%)	
Voltages: - Proportion of low voltages (<1 mV)			
	1 (0–38)%	6 (0–77)%	0.035

## Data Availability

The data that support the findings of this study are available from the corresponding author on reasonable request.

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
