# Peer review of "AF Inducibility Is Related to Conduction Abnormalities at Bachmann’s Bundle"

_jcm, 2021, doi:10.3390/jcm10235536_

Round 1

Reviewer 1 Report

Van Staveren et al investigate the potential role of Bachmann’s Bundle Conduction abnormalities and subsequent AF inducibility in 77 patients undergoing cardiac surgery. They report sustained induced AF patients:

  1. Had longer lines of conduction block
  2. Prolonged total activation times
  3. Multiple wave fronts entering Bachmann’s Bundle
  4. Greater incidence of simultaneous activation
  5. Unipolar EGM voltage reduction

These findings are interesting and the authors suggest may be of use in predicting who will develop postoperative AF.

A number of questions arise from this study

  1. It is unclear to me if the models performed included any selection between these three predictors identified in univariate analyses.  Do these predictors all overlap?

  1. Can you comment on the limitation of epicardial only mapping and an inability to assess conduction accounting for tissue thickness/endocardium?

  1. In the abstract you state, “The next step is to examine whether Bachmann’s bundle activation patterns can also be used to identify patients who will develop AF after cardiac surgery.” I believe your group published a manuscript in Circ EP in 2016 in which it is stated, “A high amount of conduction block was associated with de novo postoperative AF… long lines of longitudinal conduction block are more pronounced in patients with AF episodes.” You might want to further clarify for the reader the distinction.  I believe you are questioning if these conduction abnormalities predict AF beyond the postoperative window?  Is this correct? – If so, perhaps further clarify in the text the distinction.

Reviewer 2 Report

The manuscript by Staveren et al is well written and well presented. It describes the association of specific patterns of activation at Bachmann's bundle with AF inducibility in patients undergoing cardiac surgery. The results support the relation of conduction abnormalities at Bachmann's bundle with AF inducibility. The authors excluded patients with AF history and patients with induction of non-sustained AF. The main limitation of the study is the heterogeneous population which is inherent to the study's design (patients with defferent cardiac conditions). Moreover, the utilization of pre-existing electronic database and not a protocol designed database is another limitation. Last, the low number of inducible AF patients may lead to ambiguous statistical results, although, the authors correctly attempted to overcome this problem with the correction for BMI and LA dilatation in the multivariable analysis. The low number of events is not unexpected considering the low number of included subjects.
Besides these limitations, which, in my opinion, the authors may add to the limitations section, the study offers valuable insights for the possible association of Bachmann's bundle conduction abnormalities and AF. For sure, future, larger studies are warranted in order to confirm these results.
